# Epidemiology, Clinical, and Microbiological Characteristics of Multidrug-Resistant Gram-Negative Bacteremia in Qatar

**DOI:** 10.3390/antibiotics13040320

**Published:** 2024-03-31

**Authors:** Hamad Abdel Hadi, Soha R. Dargham, Faiha Eltayeb, Mohamed O. K. Ali, Jinan Suliman, Shiema Abdalla M. Ahmed, Ali S. Omrani, Emad Bashir Ibrahim, Yuzhou Chen, Clement K. M. Tsui, Sini Skariah, Ali Sultan

**Affiliations:** 1Communicable Diseases Centre, Hamad Medical Corporation, Doha P.O. Box 3050, Qatar; sahmed84@hamad.qa (S.A.M.A.); aomrani@hamad.qa (A.S.O.); 2College of Medicine, Qatar University, Doha P.O. Box 2713, Qatar; 3Department of Medical Education, Weill Cornell Medicine, Qatar Foundation, Doha P.O. Box 24144, Qatar; srd2003@qatar-med.cornell.edu; 4Division of Microbiology, Department of Laboratory Medicine and Pathology, Hamad Medical Corporation, Doha P.O. Box 3050, Qatar; feltayeb1@hamad.qa (F.E.); eelmagboul@hamad.qa (E.B.I.); 5Department of Internal Medicine, University Health Truman Medical Centre, Kansas City, MO 64108, USA; mabm4@umsyste.edu; 6Department of Community Medicine, Hamad Medical Corporation, Doha P.O. Box 3050, Qatar; jsuliman@hamad.qa; 7Biomedical Research Centre, Qatar University, Doha P.O. Box 2713, Qatar; 8Lee Kong Chian School of Medicine, Nanyang Technological University, Singapore 308232, Singapore; ychen114@e.ntu.edu.sg (Y.C.); clement_km_tsui@ncid.sg (C.K.M.T.); 9Infectious Diseases Research Laboratory, National Centre for Infectious Diseases, Singapore 308442, Singapore; 10Faculty of Medicine, University of British Columbia, Vancouver, BC V6T 1Z3, Canada; 11Department of Microbiology and Immunology, Weill Cornell Medicine-Qatar, Doha 2713, Qatar; sis2013@qatar-med.cornell.edu (S.S.); als2026@qatar-med.cornell.edu (A.S.)

**Keywords:** antimicrobial resistance, Gram-negative bacteria, multidrug resistance, bacteremia, bloodstream infection, Qatar

## Abstract

Antimicrobial resistance is a global healthcare threat with significant clinical and economic consequences peaking at secondary and tertiary care hospitals where multidrug-resistant Gram-negative bacteria (MDR GNB) lead to poor outcomes. A prospective study was conducted between January and December 2019 for all invasive bloodstream infections (BSIs) secondary to MDR GNB in Qatar identified during routine microbiological service to examine their clinical, microbiological, and genomic characteristics. Out of 3238 episodes of GNB BSIs, the prevalence of MDR GNB was 13% (429/3238). The predominant MDR pathogens were *Escherichia coli* (62.7%), *Klebsiella pneumoniae* (20.4%), *Salmonella* species (6.6%), and *Pseudomonas aeruginosa* (5.3%), while out of 245 clinically evaluated patients, the majority were adult males, with the elderly constituting almost one-third of the cohort and with highest observed risk for prolonged hospital stays. The risk factors identified included multiple comorbidities, recent healthcare contact, previous antimicrobial therapy, and admission to critical care. The in-hospital mortality rate was recorded at 25.7%, associated with multiple comorbidities, admission to critical care, and the acquisition of MDR *Pseudomonas aeruginosa*. Resistant pathogens demonstrated high levels of antimicrobial resistance but noticeable susceptibility to amikacin and carbapenems. Genomic analysis revealed that *Escherichia coli* ST131 and *Salmonella enterica* ST1 were the predominant clones not observed with other pathogens.

## 1. Introduction

The escalating challenges of antimicrobial resistance (AMR) is a global public health priority with significant clinical and economic outcomes [1,2,3]. Over the coming decades, evolving AMR is expected to create critical short- and long-term consequences that overshadow the outcomes from all other acute as well as chronic diseases [4,5]. To anticipate the forthcoming challenges, the scale of the problem has attracted noticeable commitment and action plans from all aspects of healthcare, including public health, direct clinical care at all levels, specialized societies, governments, and policy-makers, as well as international organizations such as the World Health Organization [6]. These joint efforts have been translated into a wider consensus to combat the problem through multiple comprehensive approaches, including global and regional surveillances strategies [6,7,8]. In frontline healthcare settings, the challenges of AMR peak in Gram-negative bacteria (GNB), since the pathogens display multiple resistance mechanisms leading to serious consequences in terms of morbidity, mortality, limited therapeutic options, increased length of hospital stay, and the increased cost of management [2,9]. Furthermore, GNB are among the foremost pathogens associated with both community- and healthcare-associated infections (HCAIs), while bloodstream infections (BSIs) constitute the ultimate clinical hazard directly linked to secondary septicemia, septic shock, admission to critical care, and eventually poor outcomes [10,11,12]. From primary infection sites, specific virulence factors in GNB allow the pathogens to propagate into BSI through the triad of direct invasion of the affected sites, overcoming the host’s defensive barriers, and survival following invasion of the bloodstream despite opposing immune defense mechanisms [13]. In secondary and tertiary care, the acquisition of GNB infection is a real threat, facilitated by the three-way links of specific pathogen-related factors such as virulence and resistance, vulnerable hosts such as patients with chronic and immunocompromised states, and the hazardous environment, such as critical care settings that facilitate accumulating risks for the transmission of HCAIs [14]. Moreover, the subsets of MDR GNB pathogens are more critical, since they are associated with poor clinical outcomes when compared with susceptible isolates, especially following delaying the initiation of appropriate antimicrobial therapy, as well as representing a serious challenge for antimicrobial stewardship and infection control and prevention programs [12,14,15,16,17].

In clinical practice, the distinction between susceptible and resistant bacterial isolates is of paramount importance, since it aids in the initial clinical evaluation and in instituting appropriate therapy, as well as quantifying an assessment of the risks for their safe clinical management [18]. Consequently, most prominent specialized international societies regularly advocate and update appropriate evaluations to direct management specifically aimed at MDR pathogens [19,20].

Despite the importance of the problem from global and local perspectives at all levels, there is a paucity of data specifically from our region of MDR GNB manifesting as BSIs [21]. To bridge such gaps, experts advocate strengthening surveillance methods to comprehend the scale of the problem and to evaluate its different local aspects, including the microbiological and clinical characteristics [22]. The present study aimed to prospectively evaluate the epidemiology, clinical, and microbiological characteristics of bloodstream infections caused by MDR GNB from Qatar which started in 2019.

## 2. Materials and Methods

### 2.1. Settings

Hamad Medical Corporation (HMC) is the main healthcare provider in the country, acting through 14 acute and specialized hospitals with a total capacity of almost 2500 beds that serve the country’s population of almost three million. In HMC, there is a national electronic health information system that allows for the retrieval of data on patients and pathogens [23]. The microbiology division of the department of pathology and laboratory medicine handles all specimens for microbiological identification, antimicrobial susceptibility tests (AST), and reporting through a major central facility and few peripherals. The laboratory is equipped with modern electronic and diagnostic equipment that complies with international standards and receives regular inspections and accreditation [24].

### 2.2. Epidemiology and Clinical Demographics

For the study year of 2019, the electronic records of patients from the main general hospital identified with positive BSI were reviewed to include the patients’ demographics, risk factors, evaluation, and management including the severity of the infection as well as the microbiological and genomic characteristics. In total, 386 non-duplicate cases of BSI were identified in the year 2019; out of these, clinical details and antibiotic susceptibility tests were retrieved for 245 cases hosted in the central HMC database. The cascade of the study flow chart is outlined in Appendix A.

### 2.3. Bacterial Identification and Antimicrobial Susceptibility Testing (ASTs)

All isolates were collected from positive blood culture specimens from all age groups received at the central and peripheral microbiology laboratories across the HMC’s sites. Microbiological identification and ASTs were performed using automated MALDI-TOF spectrometry (Bruker Daltonics, Billerica, MA, USA) and BD Phoenix™ Microbiology Systems (BD Diagnostics, Durham, NC, USA). Interpretation of the susceptibility results was based on the breakpoints of the Clinical Laboratory Standards Institute (CLSI), along with HMC’s internal institutional guidelines [25]. Consecutive BSIs were prospectively reported to identify clinically relevant GNB, which were furtherly subclassified into MDR on the basis of their resistance to at least one antimicrobial agent from three different antimicrobial classes, as widely agreed, while ESBLs were identified with a MIC of >2 µg/mL for third-generation cephalosporins and confirmed by double-disc diffusion methods [25,26].

### 2.4. Molecular and Genomic Evaluation through WGS Methods

For whole-genome sequencing, MDR-GNB BSIs with available AST data were ranked on the basis of their decreasing resistance against the total number of antibiotics tested, and the top 100 resistant isolates were selected on the basis of budget constraints; for example, all carbapenem-resistant (CR) GNB were included. Genomic DNA was extracted using a DNeasy Blood and Tissue Kit (Qiagen, Germany), and the DNA libraries were sequenced on the Illumina NextSeq 550 platform using 2 × 150 bp PE at SeqCenter (Pittsburgh, PA, USA) [27]. The raw reads were assembled de novo using SPAdes v.3.9.0 implemented in shovill (https://github.com/tseemann/shovill (accessed on 1 March 2024)) [28,29]. Sequence types, (ST), plasmid replicons, and AMR genes were predicted from the assembled contigs using the multilocus sequence typing (MLST) (https://github.com/tseemann/mlst (accessed on 1 March 2024)), Plasmidfinder v2.1, and ResFinder v3.2 databases implemented in ABRicate v0.9 (https://github.com/tseemann/abricate (accessed on 1 March 2024)), on the basis of >70% coverage and 90% sequence identity [30]. Previously unreported *Klebsiella species* ST were submitted to BIGSDB (https://bigsdb.pasteur.fr/klebsiella/ (accessed on 1 March 2024)) for ST assignment.

### 2.5. Statistical Analysis

Analysis of the data was conducted using IBM-SPSS version 27.0 (Armonk, NY, USA). The samples’ characteristics were summarized using frequency and percentage distributions, while Fisher’s exact and Chi-square tests were conducted to identify the bivariate associations of the duration of hospital stay and in-hospital mortality. To examine for the influence of different variables on the outcomes, a multivariable regression model was performed using the variables of age, gender, pathogens, admission to critical care, and duration of hospital stay, and records of comorbidity scores were examined against the outcome of in-hospital mortality. Furthermore, collinearity was tested among the categorical independent variables using Chi-square and/or Fisher’s exact tests, with the estimates of the adjusted odds ratio (OR) and confidence intervals set at 95%. Covariates with *p*-values of ≤0.05 in the multivariable analysis were regarded as strong evidence for associations.

## 3. Results

During the year 2019, out 3238 episodes of Gram-negative bloodstream infections, 429 fulfilled the criteria for multidrug resistance (MDR), with a prevalence of 13% (429/3238); of these, 386 were non-duplicates isolates representing monomicrobial infections in 95% of cases and polymicrobial infections in 5%. The key identified pathogens were *E. coli* (62.7%), *Klebsiella* species (20.4%), *Salmonella* species (6.6%), and *P. aeruginosa* (5.3%), while others were 5% (comprising *Klebsiella* species other than *Klebsiella pneumoniae*, *Enterobacter cloacae*, *Serratia marcescens*, *Bacteroides fragilis*, *Citrobacter freundii*, *Acinetobacter baumannii*, and *Proteus mirabilis*) (Table 1). In the evaluation of 245 patients enrolled for the first year of the study, most patients were adults (98.4%) and males (61.6%), while the elderly aged >65 years, who had the highest risk of prolonged hospital stays, constituted 36.6% (Table 1). The majority of patients had multiple comorbidities, dominated by diabetes mellitus (46.6%), coronary heart diseases (29.4%), neoplastic diseases (27.8%), and chronic and end-stage kidney diseases (26.6%), who were admitted to or in critical care settings during the study period in 73.8% of cases and were treated with >2 antimicrobials in more than half of the cases (51.4%). Risk factors for the acquisition of invasive MDR included previous healthcare contact (67.8%) and previous antimicrobial exposure in the preceding 90 days. The major sources of BSI were the urinary tract (45.5%), intrabdominal infections (19.6%), and the related central lines (10.2%), with an in-hospital mortality rate of 25.7% (Table 2). In almost 50% of the cohorts, the patients were exposed to different classes of antimicrobials as therapy before and during their hospital admission preceding the isolation of MDR GNB BSI. Following that, carbapenems and aminoglycosides were the main directed therapy for the treatment of MDR GNB BSI (Figure 1).

Microbiological evaluation of 245 isolates revealed a high level of resistance to penicillin’s (100%), cephalosporins (75–100%), and quinolones (41.7–68.6%) as well as moderate resistance to β-lactam β-lactamase inhibitors (BLBLIs) (11.3–15.7%) compared with high antimicrobial susceptibility to aminoglycosides and carbapenems (74.2–100%) (Table 3).

The genomic characterization of the 100 selected isolates displaying the highest antimicrobial resistance revealed that the most frequent pathogens were *E. coli* (*n* = 38) and *K. pneumoniae* (*n* = 34), followed by *P. aeruginosa* (*n* = 12) and *Salmonella enterica* (*n* = 10). Isolates of *E. coli* belonged to 16 different sequence types (ST), and ST131 (*n* = 13) was predominant, followed by ST167 (*n* = 4) and ST410 (*n* = 4). *K. pneumoniae* was represented by 26 different STs, and the most frequent one was ST16 (*n* = 3), with no specific clustering. Similarly, all 12 *P. aeruginosa* isolates belonged to different STs, in contrast to all *Salmonella enterica* samples, which belonged to ST1 (*n* = 10). According to the genomic evaluation, *E. coli* and *Salmonella enterica* causing BSI appeared to exhibit a clonal propensity (Table 4). To distinguish correlated factors associated with in-hospital mortality, we performed a regression analysis, and identified multiple comorbidities (OR 17.38 CI: 2.4–123.8), admission to critical care (OR 6.67, CI: 3–16.7), and infections secondary to MDR *P. aeruginosa* (OR 13.9, CI: 2.3–82.8) as having significant associations with poor outcomes (Table 5).

## 4. Discussion

The impact of antimicrobial resistance (AMR) is a global healthcare challenge because of the direct clinical outcomes in terms of morbidity and mortality, as well as indirect sequelae such as the economic and social consequences [2,3]. Amongst the spectrum of AMR pathogens, Gram-negative bacteria (GNB) represent a prominent aspect, since they possess unique resistance mechanisms allowing them to exhibit phenotypic multidrug resistance (MDR) profiles rendering them resistant to conventional and advanced antimicrobials with limited therapeutic options [4,31]. Furthermore, the pathogens represent a leading cause of community- and hospital-acquired infections (HAIs) such as urinary tract and intrabdominal infections, as well as HAIs such as pneumonia, urinary, and bloodstream infections (BSIs). Invasive GNB pathogens, particularly MDR BSIs, constitute a pivotal category, since they are almost always pathological, with frequent catastrophic clinical consequences such as septicemia, septic shock, and multiorgan dysfunctions, frequently leading to admission to critical care and with the highest mortality rate when compared with other pathogens [13,18,19].

Our evaluation of MDR GNB bacteremia highlighted an overall prevalence of 13% of all invasive episodes. Although the overall prevalence of MDR GNB bacteremia is similar to other regions, it is certainly higher for *Enterobacterales* [32]. The key top identified pathogens were *E. coli*, *Klebsiella* species, *Salmonella* species, *P. aeruginosa*, *Serratia marcescens*, and *Enterobacter cloacae*, constituting 98% of all MDR pathogens. While three out of the four key pathogens are commonly reported in different parts around the world, MDR *Salmonella* infection is certainly a regional phenomenon reflecting imported travel-related infections from the neighboring Indian subcontinent and Africa, where resistant *Salmonella* infections are endemic and frequently reported [32,33]. This established link of imported MDR *Salmonella* bacteremia was observed in the country almost three decades ago, and hence it has remained a substantial management challenge [32,33]. In multiple similar surveillance studies, *Salmonella* BSI has been one of the leading causes of bacteremia in developing countries, particularly in Asia and Africa [34]. Distinctively, although MDR *Acinetobacter baumanaii* bacteremia has been reported as a leading entity in different global regions, including North America, South America, and Europe, it remains rare in our region [32,33,34,35,36]. Despite the establishment of the clonal endemicity of *Acinetobacter baumanaii* in our clinical settings, as documented in a previous study almost a decade previously, there was a paucity of the related invasive diseases, with a predominance of those of respiratory origin, which might suggest the absence of virulent blood-invasive clones [37,38]. 

Of note, although the study encompassed all age groups, it is noticeable that the majority of cases were adults (98%), pointing towards the observation that MDR GNB bacteremia was rare in the examined children, with supporting additional evidence that the majority of patients had multiple comorbidities, multiple previous encounters with healthcare facilities, and exposure to previous antimicrobial therapy, conditions that are usually associated with older age groups. These risk factors for the acquisition MDR GNB including BSI have been reported before, pointing towards cumulative evolutionary shift of GNB towards resistance [17,36,39,40]. Since the country’s demographics have a young male distribution, the gender difference was probably not significant, but the high proportion of affected elderly patients was certainly related. The fact that the elderly (>65 years of age) constituted more than one-third of the cohort is certainly significant, coupled with noticeable prolonged hospital stays of >90 days [36]. Old-age and residence in long-term facilities have been described before as being prime risk factors for the acquisition of MDR GNB bacteremia and are similarly associated with prolonged hospital stays as well as being independent risk factors of increased mortality [36,41,42,43]. 

Regarding the risk factors for the acquisition of MDR BSI, patients with multiple comorbidities, particularly immune suppression, and neoplastic diseases, have been linked to infection with MDR GNB [44]. Likewise, the acquisition of invasive MDR GNB has been associated with previous exposure to antimicrobial therapy and previous admission to healthcare facilities, leading to prolonged hospital stays [36,45]. It must be highlighted that the observed in-hospital mortality rate for MDR GNB BSI was recorded as 25.7%, which is similar to that in other similar regional studies, although some recorded high mortality rates exceeding 50% that were linked to MDR *K. pneumoniae* and *Acinetobacter baumannii* [46,47,48]. The predominance of MDR *E. coli*, coupled with a paucity of resistant *Acinetobacter baumannii* in our cohort probably contributed to the lower comparable mortality rates. Additionally, examined risk factors correlating with mortality, namely age, gender, and duration of hospital stay, were not significant observed risk factors, while multiple and high comorbidity scores, admission to critical care, and acquisition of *P. aeruginosa* had strong associations.

For the phenotypic review, the microbiological evaluation highlighted the prominent resistance (98–100%) of MDR GNB to all penicillin’s, monobactams such as aztreonam, and all classes of cephalosporins (except for cefoxitin), including fourth-generation agents, represented by cefepime. This was an expected finding, since the prime mechanism of resistance in MDR GNB is the production of extended spectrum β-lactamases (ESBL) that inactivate almost all penicillin-based antimicrobials, including monobactams and cephalosporins [18,49,50]. Because carbapenems such as meropenem, imipenem, and ertapenem are resistant to the inactivation of ESBL to the point of being considered the sine qua non for the management for ESBL-producing GNB, particularly invasive diseases, intriguingly, the results demonstrated noticeable overall differences as well as pathogen-specific activity profiles [51]. For example, the AST for the three highlighted carbapenem agents against *E. coli* was 99.4%, 95.6%, and 93.1%, respectively, while those for *Klebsiella* species were 76.3%, 72.9%, and 72.9%, respectively. This can be partially explained by the prominent production of ESBL-resistant genes by *E. coli* when compared with *Klebsiella* species, which tend to display more advanced resistance mechanisms, such as the genotypic production of carbapenem-resistant genes (CRGs) that are capable of inactivating carbapenems [52,53]. Since the displayed results highlighted a high level of resistance in *Klebsiella* species against carbapenems, further evaluations revealed they were true CREs. Nevertheless, the conspicuous resistance pattern of *Klebsiella* species compared with *E. coli* was observed in other similar surveillance studies [52]. Moreover, as well as the production of ESBL and CRGs conferring MDR, GNB also produce other non-enzymatic resistance mechanisms, such as porins and efflux pump mutations that impede antimicrobials’ entry and efflux into the targeted pathogens, leading to phenotypic resistance pattens. Since there is a non-uniform susceptibility of different carbapenems to these mechanisms, most likely it accounts for the inter-class variations [51,54]. Despite that, carbapenems were the main selected therapeutic agents used for the management of invasive MDR GNB for the cohort. Comparatively, advanced BLBLIs represented by piperacillin–tazobactam demonstrated suboptimal activity (85.5%) against MDR *E. coli* and 44.1% against *Klebsiella*, with poor activity against MDR *P. aeruginosa* (8.3%). The historic position for piperacillin–tazobactam in treating MDR GNB bacteremia, particularly for *Enterobacterales*, was severely dented when a large randomized clinical trial demonstrated that it was inferior to carbapenems, with increased mortality [55]. Despite concerns raised over the use of BLBLIs for the treatment of invasive MDR GNB such as ESBL BSI, nevertheless, a collective analysis of retrospective clinical studies showed no observed differences, while ongoing clinical trials are still exploring this contentious concept [56,57]. Similarly, there were noticeable inter-class variations between the two main aminoglycosides, namely amikacin and gentamycin, as the former demonstrated wider AST activity against MDR GNB spanning *E. coli* (100%), *Klebsiella* species (91.5%), and *P. aeruginosa* (91.7%) compared with 74.2%, 69.5%, and 75%, respectively, for gentamycin. Despite the historic use of this antimicrobial class in clinical practice for almost 80 years, there is a paucity of data regarding comparative studies between the different agents, particularly for MDR GNB, to guide clinical care, an area that we recommend being explored further [58,59,60,61]. 

In MDR GNB, the hallmark for resistance to aminoglycosides is enzymatic inactivation or modifications such as aminoglycoside-modifying enzymes (AMEs), which are readily expressed by many MDR GNB [59]. In the previous limited studies, amikacin’s ASTs were usually higher than those of gentamycin against MDR GNB, despite the cautious calls for the accurate interpretation of CLSI and EUCAST breakpoints for amikacins, especially during co-occurrence of AMEs with CRGs [62,63]. Additionally, *Salmonella* species, particularly resistant strains, have poor permeability to aminoglycosides, reflected by the universal resistance in our cohort (100%) [64]. 

It is certainly worth highlighting the high-level resistance of the MDR GNB in our cohort to quinolones. For *E. coli*, identical resistance to ciprofloxacin and levofloxacin was found (68.6%), while for *Klebsiella* species, it was 80.8% and 64.1%, respectively. Quinolones represent an attractive therapeutic alternative for GNB because of their unique mode of action, inhibiting DNA synthesis but hindered by the escalating resistance in MDR isolates, which started almost two decades ago [65]. This usually occurs because of the frequent co-transmission of plasmid-mediated quinolone-resistance genes (QRGs) that impede its activity [66]. In GNB bacteremia, quinolone resistance has been shown to be associated with MDR for related pathogens, as well as an independent risk factor for prolonged hospital stays [67]. This can be challenging during the management of invasive disease; for example, although quinolones are classically associated with the management of *Salmonella* infections, all the cohort’s isolates were resistant. 

From the review of the microbiological phenotypic characteristics, it must be highlighted that certain pathogens possess intrinsic mechanisms of resistance that render the tested or used antimicrobials ineffective, such as the historic association between *K. pneumoniae* and resistance to ampicillin, *Enterobacter cloacae* and *Serratia marcescens* having resistance to most tested penicillin and cephalosporins, and the similar recognized specific resistance of *P. aeruginosa* to ertapenem and tigecycline. Conversely, the paucity of BSI caused by *Acinetobacter baumanii*, *Enterobacter cloacae*, and *Serratia marcescens* limited accurate comparisons of antimicrobial activity relating to these pathogens.

Lastly, for MDR pathogens, the genomic evaluation revealed the predominance of sequence types ST131 and ST167 for *E. coli*, and ST410 and ST1 for *Salmonella* species. *E. coli* ST131 is a global MDR clone associated with both community-associated infections and HAIs, with a multitude of resistance genes, including ESBLs, particularly *bla_CXM-15_*, and quinolone resistance genes and virulence factors specifically associated with UTIs and bacteremia [68,69]. For *E. coli*, the results also reported ST167 and ST410, which are emerging global MDR bacterial clones associated with carbapenem resistance [70,71].

Over the last decades, the sequential accumulation of multiple plasmid-mediated resistance genes has conferred resistance almost to all penicillin, cephalosporins, and quinolones [72]. These observations are highlighted in our study, where the majority of BSI were from urinary sources, and resistance to penicillin, cephalosporins, and quinolones were evidently pronounced for *E. coli* Similarly, the epidemic and resistant clone of *Salmonella* ST1 is related to invasive enteric *S. typhi*, and is associated with spreading MDR in different endemic regions around the world, including the Indian subcontinent, as reported in our cohort [73,74,75]. Intriguingly, despite previous studies from our institution highlighting circulating endemic clones for hypervirulent and MDR *Klebsiella* species such as ST383, genomic characterization of the invasive *Klebsiella* BSI did not reveal such clustering patterns [76].

Despite this national study being the largest and most comprehensive surveillance for MDR GNB bacteremia in the region and one of few globally, it has some limitations. For logistic reasons, reported clinical and microbiological data were only available for two-thirds of all identified bloodstream infections. However, the results are in line with other reported bacteremia studies, which support their validity. Additionally, only about one-quarter of the isolates were processed for molecular studies, limited by budget constraints, although comprehensive evaluations certainly would have added to our understanding of the invasive disease. Nevertheless, the study provides a new benchmark in epidemiology, clinical profiles, and antimicrobial characteristics, which will be a reference for the country and the region for the important and well-defined condition of invasive bloodstream infection.

In conclusion, this large prospective study of invasive MDR GNB associated with BSI from Qatar demonstrated significant multidrug resistance. Adults and elderly patients are more likely to be affected by the invasive disease, usually those suffering from multiple clinical comorbidities linked to admission to critical care. The microbiological characteristics and phenotypic profiles demonstrated that MDR *E. coli*, *Klebsiella* species, *Salmonella* species, and *P. aeruginosa* are the main pathogens with high-level resistance to most antimicrobials and prime susceptibility to carbapenem and amikacin. Clonal clustering was observed for *E. coli* and *Salmonella enterica*, but not for the other pathogens.

## Figures and Tables

**Figure 1 antibiotics-13-00320-f001:**
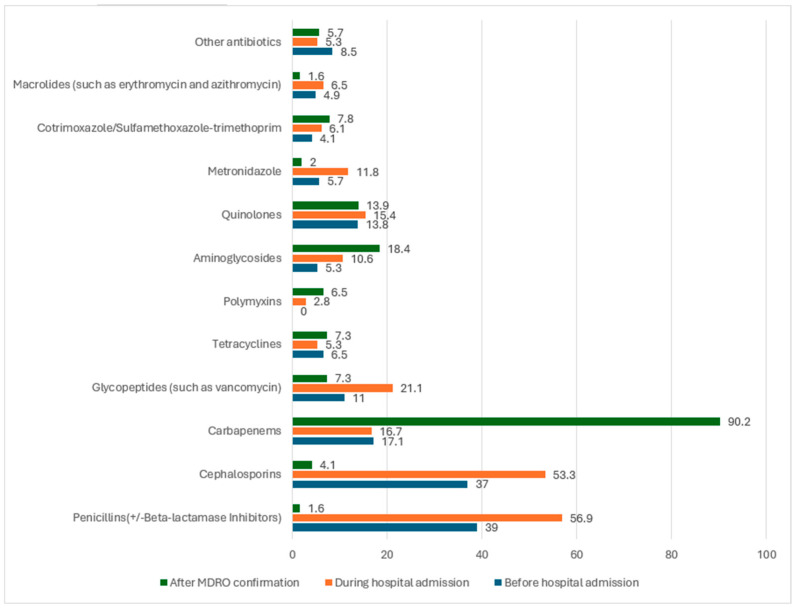
Graph showing the antibiotic treatment profile of 245 patients with multidrug-resistant invasive Gram-negative infections collected in the current study. For each antimicrobial class, the blue bars represent antimicrobials received before hospital admission and orange are those received during hospital admission before the isolation of the resistant pathogens, while the green are those received following the isolation of MDR GNB bacteremia. Carbapenems and aminoglycosides were the main classes used for targeted therapy.

**Table 1 antibiotics-13-00320-t001:** Frequency and percentage of multidrug-resistant invasive Gram-negative organisms. The incidence of individual organisms includes both single (95%) and polymicrobial infections (5%) among the 386 non-duplicate cases of invasive MDR GNB infections identified in the year 2019.

Organism	N	%
*Escherichia coli*	249	62.7
*Klebsiella pneumoniae*	81	20.4
*Salmonella Typhi*	26	6.6
*Pseudomonas aeruginosa*	21	5.3
** Others*	20	5
**Total**		**100**

* *Klebsiella* species other than *Klebsiella pneumoniae*, *Enterobacter cloacae*, *Serratia marcescens*, *Bacteroides fragilis*, *Citrobacter freundii*, *Acinetobacter baumannii*, and *Proteus mirabilis*.

**Table 2 antibiotics-13-00320-t002:** Demographic profile of 245 patients with invasive multidrug resistant Gram-negative bacterial bloodstream infections collected during 2019.

Demographics		Total	Duration of Hospital Stay			*p*-Value
			Short Stay	Medium Stay	Long Stay	
		N (%)	(<30 Days) N (%)	(30–90 Days) N (%)	(90+ Days) N (%)	
**Age**	**Pediatric (<14 years)**	4 (1.7)	3 (1.9)	0 (0.0)	1 (2.4)	0.121 ^a^
	**Adult (14–65 years)**	147 (61.8)	99 (64.3)	29 (69.0)	19 (45.2)	
	**Geriatric (>65 years)**	87 (36.6)	52 (33.8)	13 (31.0)	22 (52.4)	
**Gender**	**Male**	151 (61.6)	93 (58.1)	33 (78.6)	25 (58.1)	0.047 ^b^
	**Female**	94 (38.4)	67 (41.9)	9 (21.4)	18 (41.9)	
**ICU stay**	**Present**	180 (73.8)	131 (81.9)	21 (50.0)	28 (66.7)	<0.001 ^b^
**Source**	**HAP ^c^**	6 (2.4)	4 (2.5)	0 (0.0)	2 (4.7)	<0.001 ^a^
	**VAP ^d^**	8 (3.3)	1 (0.6)	4 (9.5)	3 (7.0)	
	**CAUTI ^e^**	11 (4.5)	4 (2.5)	4 (9.5)	3 (7.0)	
	**CLABSI ^f^**	21 (8.6)	3 (1.9)	10 (23.8)	8 (18.6)	
	**Other**	199 (81.2)	148 (92.5)	24 (57.1)	27 (62.8)	
**Acquisition**	**HCAI**	89 (37.9)	29 (18.8)	28 (70.0)	32 (78.0)	<0.001 ^b^
**Extensive health care contact**	**Present**	166 (67.8)	86 (53.8)	39 (92.9)	41 (95.3)	<0.001 ^b^
**Readmission within 28 days**	**Present**	36 (14.8)	25 (15.8)	8 (19.0)	3 (7.0)	0.259 ^b^
**Comorbidities**	**DM**	118 (48.6)	72 (45.6)	21 (50.0)	25 (58.1)	0.343 ^b^
	**Heart disease/heart failure/CHD**	72 (29.4)	38 (23.8)	21 (50.0)	13 (30.2)	0.004 ^b^
	**Chronic liver disease/biliary disease**	27 (11.0)	18 (11.3)	3 (7.1)	6 (14.0)	0.572 ^a^
	**CKD/ESRD**	65 (26.6)	36 (22.6)	11 (26.2)	18 (41.9)	0.040 ^b^
	**Malignancy**	68 (27.8)	34 (21.3)	19 (45.2)	15 (34.9)	0.004 ^b^
	**Post-transplantation**	9 (3.7)	4 (2.5)	1 (2.4)	4 (9.3)	0.115 ^a^
	**Neutropenic**	25 (10.2)	10 (6.3)	9 (21.4)	6 (14.0)	0.009 ^a^
	**Cystic fibrosis**	0 (0.0)	0 (0.0)	0 (0.0)	0 (0.0)	N/A
	**Renal stones/urinary tract abnormality**	23 (9.4)	19 (11.9)	2 (4.8)	2 (4.7)	0.270 ^a^
	**Chronic lung disease**	24 (9.8)	15 (9.4)	5 (11.9)	4 (9.3)	0.818 ^a^
**Invasive devices**	**Central line**	83 (34.0)	27 (17.0)	31 (73.8)	25 (58.1)	<0.001 ^b^
	**Foly’s d**	64 (26.3)	25 (15.7)	17 (41.5)	22 (51.2)	<0.001 ^a^
	**NGT d**	46 (18.9)	9 (5.7)	16 (38.1)	21 (48.8)	<0.001 ^a^
	**DJ stent/PCN**	5 (2.0)	4 (2.5)	1 (2.4)	0 (0.0)	0.829 ^a^
	**Mechanical ventilator (intubated)**	38 (15.5)	8 (5.0)	12 (28.6)	18 (41.9)	<0.001 ^a^
	**Tracheostomy tube**	18 (7.3)	1 (0.6)	6 (14.3)	11 (25.6)	<0.001 ^a^
	**PEG tube**	5 (2.0)	1 (0.6)	0 (0.0)	4 (9.3)	0.007 ^a^
	**Other invasive device**	28 (11.5)	17 (10.7)	6 (14.3)	5 (11.6)	0.385 ^a^
**Number of antibiotics**	**0**	8 (3.3)	8 (5.0)	0 (0.0)	0 (0.0)	0.023 ^a^
	**1**	11 (45.3)	79 (49.4)	16 (38.1)	16 (37.2)	
	**2+**	126 (51.4)	73 (45.6)	26 (61.9)	27 (62.8)	
**Source of bacteremia**	**Present**	206 (84.1)	131 (81.9)	37 (88.1)	38 (88.4)	0.444 ^b^
	**Intraabdominal**	48 (19.6)	36 (22.5)	8 (19)	4 (9.3)	0.154 ^b^
	**Urinary system**	111 (45.5)	89 (56)	12 (28.6)	10 (23.3)	<0.001 ^b^
	**Skin and soft tissue**	22 (9.0)	5 (3.1)	6 (14.3)	11 (25.6)	<0.001 ^a^
	**Line related**	25 (10.2)	3 (1.9)	10 (24.4)	12 (27.9)	<0.001 ^a^
	**CNS**	2 (0.8)	0 (0)	0 (0)	2 (4.7)	0.060 ^a^
	**Other sites**	7 (2.9)	3 (1.9)	2 (4.8)	2 (4.7)	0.305 ^a^

^a^ *p*-value estimated using Fisher’s exact test; ^b^ *p*-value estimated using the Chi-square test. ^c^ HAP, hospital-acquired pneumonia; ^d^ VAP, ventilation-associated pneumonia; ^e^ CAUTI, catheter-associated urinary tract infection; ^f^ CLABSI, central line-associated bloodstream infection; HCAIs, healthcare-associated infections; DM, diabetes mellitus; CHD, coronary heart disease; CKD, chronic kidney disease; ESRD, end-stage renal disease; NGT, nasogastric tube; PCN, percutaneous nephrostomy; PEG, percutaneous endoscopic gastrostomy; CNC, central nervous system.

**Table 3 antibiotics-13-00320-t003:** Antimicrobial susceptibility test (AST) results of 245 invasive multidrug-resistant Gram-negative bacteria isolates collected during 2019.

Antibiotic	Interpretation of Resistance	*Escherichia coli*	*Klebsiella pneumoniae*/spp.	*Pseudomonas aeruginosa*	*Salmonella enterica* ssp./*Salmonella* sp.	*Enterobacter cloacae*	*Serratia marcescens*
		N (%)	N (%)	N (%)	N (%)	N (%)	N (%)
**Amikacin**	**S**	159 (100)	54 (91.5)	11 (91.7)	0 (0)	3 (100)	2 (100)
	**R**	0 (0)	5 (8.5)	1 (8.3)	10 (100)	0 (0)	0 (0)
	**I**	0 (0)	0 (0)	0 (0)	0 (0)	0 (0)	0 (0)
**Cefoxitin**	**S**	125 (78.5)	35 (59.3)		0 (0)		
	**R**	23 (14.5)	21 (35.6)	* NA	10 (100)	* NA	* NA
	**I**	11 (7)	3 (5.1)		0 (0)		
**Ertapenem**	**S**	148 (93.1)	43 (72.9)		10 (100)	3 (100)	0 (0)
	**R**	7 (4.4)	15 (25.4)	* NA	0 (0)	0 (0)	2 (100)
	**I**	4 (2.5)	1 (1.7)		0 (0)	0 (0)	0 (0)
**Imipenem**	**S**	152 (95.6)	43 (72.9)	0 (0)	10 (100)	3 (100)	0 (0)
	**R**	1 (0.6)	15 (25.4)	12 (100)	0 (0)	0 (0)	2 (100)
	**I**	6 (3.8)	1 (1.7)	0 (0)	0 (0)	0 (0)	0 (0)
**Meropenem**	**S**	158 (99.4)	45 (76.3)	0 (0)	10 (100)	3 (100)	0 (0)
	**R**	1 (0.6)	14 (23.7)	11 (91.7)	0 (0)	0 (0)	2 (100)
	**I**	0 (0)	0 (0)	1 (8.3)	0 (0)	0 (0)	0 (0)
**Nitrofurantoin**	**S**	149 (93.7)	17 (28.8)		10 (100)	0 (0)	
	**R**	5 (3.1)	27 (45.8)	* NA	0 (0)	3 (100)	* NA
	**I**	5 (3.1)	15 (25.4)		0 (0)	0 (0)	
**Tigecycline**	**S**	158 (99.4)	22 (37.3)		10 (100)	3 (100)	0 (0)
	**R**	1 (0.6)	15 (25.4)	* NA	0 (0)	0 (0)	0 (0)
	**I**	0 (0)	22 (37.3)		0 (0)	0 (0)	2 (100)
**Amoxicillin clavulanate**	**S**	88 (55.3)	19 (32.2)		10 (100)		
	**R**	25 (15.7)	23 (39)	* NA	0 (0)	* NA	* NA
	**I**	46 (28.9)	17 (28.8)		0 (0)		
**Ampicillin**	**S**	0 (0)			0 (0)		
	**R**	159 (100)	* NA	* NA	10 (100)	* NA	* NA
	**I**	0 (0)			0 (0)		
**Aztreonam**	**S**	3 (1.9)	3 (5.1)	0 (0)	0 (0)	0 (0)	2 (100)
	**R**	156 (98.1)	56 (94.9)	12 (100)	10 (100)	3 (100)	0 (0)
	**I**	0 (0)	0 (0)	0 (0)	0 (0)	0 (0)	0 (0)
**Cefazolin**	**S**	0 (0)	0 (0)		0 (0)		
	**R**	159 (100)	59 (100)	* NA	10 (100)	* NA	* NA
	**I**	0 (0)	0 (0)		0 (0)		
**Cefepime**	**S**	2 (1.3)	0 (0)	0 (0)	0 (0)	1 (33.3)	1 (50)
	**R**	157 (98.7)	58 (98.3)	12 (100)	10 (100)	2 (66.7)	1 (50)
	**I**	0 (0)	1 (1.7)	0 (0)	0 (0)	0 (0)	0 (0)
**Ceftazidime**	**S**	3 (1.9)	2 (3.4)	1 (8.3)	0 (0)	0 (0)	2 (100)
	**R**	156 (98.1)	57 (96.6)	9 (75)	10 (100)	3 (100)	0 (0)
	**I**	0 (0)	0 (0)	2 (16.7)	0 (0)	0 (0)	0 (0)
**Ceftriaxone**	**S**	3 (1.9)	1 (1.7)		0 (0)	0 (0)	0 (0)
	**R**	156 (98.1)	57 (96.6)	* NA	10 (100)	3 (100)	2 (100)
	**I**	0 (0)	1 (1.7)		0 (0)	0 (0)	0 (0)
**Cefuroxime**	**S**	2 (1.3)	0 (0)		0 (0)		0 (0)
	**R**	157 (98.7)	59 (100)	* NA	10 (100)	* NA	* NA
	**I**	0 (0)	0 (0)		0 (0)		0 (0)
**Cephalothin**	**S**	0 (0)	0 (0)		0 (0)		0 (0)
	**R**	158 (99.4)	59 (100)	* NA)	10 (100)	* NA	* NA
	**I**	1 (0.6)	0 (0)		0 (0)		0 (0)
**Ciprofloxacin**	**S**	48 (30.2)	8 (15.4)	7 (58.3)	0 (0)	1 (50)	2 (100)
	**R**	109 (68.6)	42 (80.8)	4 (33.3)	9 (100)	1 (50)	0 (0)
	**I**	2 (1.2)	2 (3.8)	1 (8.3)	0 (0)	0 (0)	0 (0)
**Gentamicin**	**S**	118 (74.2)	41 (69.5)	9 (75)	0 (0)	2 (66.7)	2 (100)
	**R**	40 (25.2)	18 (30.5)	1 (8.3)	10 (100)	1 (33.3)	0 (0)
	**I**	1 (0.6)	0 (0)	2 (16.7)	0 (0)	0 (0)	0 (0)
**Levofloxacin**	**S**	49 (30.8)	12 (30.8)	4 (33.3)	0 (0)	1 (100)	2 (100)
	**R**	109 (68.6)	25 (64.1)	5 (41.7)	7 (100)	0 (0)	0 (0)
	**I**	1 (0.6)	2 (5.1)	3 (25)	0 (0)	0 (0)	0 (0)
**Piperacillin–tazobactam**	**S**	136 (85.5)	26 (44.1)	1 (8.3)	0 (0)	3 (100)	0 (0)
	**R**	18 (11.3)	28 (47.5)	9 (75)	10 (100)	0 (0)	2 (100)
	**I**	5 (3.1)	5 (8.5)	2 (16.7)	0 (0)	0 (0)	0 (0)
**Trimethoprim sulfamethoxazole**	**S**	53 (33.3)	11 (18.6)		0 (0)	3 (100)	2 (100)
	**R**	106 (66.7)	48 (81.4)	* NA	10 (100)	0 (0)	0 (0)
	**I**	0 (0)	0 (0)		0 (0)	0 (0)	0 (0)
**Colistin**	**S**	1 (100)	0 (0)	0 (0)	0 (0)	0 (0)	
	**R**	0 (0)	2 (100)	0 (0)	0 (0)	1 (100)	* NA
	**I**	0 (0)	0 (0)	0 (0)	0 (0)	0 (0)	
**Polymyxin**	**S**	0 (0)	0 (0)	0 (0)	0 (0)	0 (0)	
	**R**	0 (0)	0 (0)	0 (0)	0 (0)	0 (0)	* NA
	**I**	0 (0)	0 (0)	0 (0)	0 (0)	0 (0)	

* NA: These organisms are intrinsically resistant to the designated antimicrobials, and their AST results should be interpreted as categorically resistant or ineffective.

**Table 4 antibiotics-13-00320-t004:** Regression analysis model identifying risk factors correlating with in-hospital mortality for invasive MDR Gram-negative infections.

		Death	
		OR (95% CI)	*p*-Value
**Age**	Adult (14–65 years)	Ref.	
	Geriatric (>65 years)	2.01 (0.94–4.3)	0.073
**Sex**	Male	Ref.	
	Female	1.23 (0.56–2.66)	0.606
**ICU stay**	No	Ref.	
	Yes	6.67 (3.03–16.67)	<0.001
**Duration of stay**	Short stay (<30 days)	Ref.	
	Medium stay (30–90 days)	0.51 (0.18–1.49)	0.221
	Long stay (90+ days)	1.29 (0.48–3.52)	0.613
**Comorbidity score**	0	Ref.	
	1	4.76 (0.69–32.59)	0.112
	2	9.62 (1.52–60.69)	0.016
	3	9.53 (1.37–66.56)	0.023
	4+	17.38 (2.44–123.79)	0.004
**Organism**	*Escherichia coli*	Ref.	
	*Klebsiella*	2.54 (1.1–5.88)	0.029
	*Pseudomonas aeruginosa*	13.94 (2.35–82.81)	0.004

**Table 5 antibiotics-13-00320-t005:** Multilocus sequence typing results of 100 selected invasive multidrug resistant Gram-negative organisms collected in the current study.

Organism	ST	N	(%)	Organism	ST	N	(%)
*Escherichia coli*	10	1	1	*Klebsiella aerogenes*	ND	1	1
	38	1	1	*Klebsiella oxytoca*	86	1	1
	131	13	13		194	1	1
	167	4	4				
	224	1	1	*Pseudomonas aeruginosa*	16	1	1
	361	2	2		132	1	1
	405	1	1		234	1	1
	410	4	4		235	1	1
	450	1	1		244	1	1
	617	2	2		260	1	1
	648	2	2		500	1	1
	1193	2	2		1417	1	1
	2083	1	1		1992	1	1
	2279	1	1		3045	1	1
	2345	1	1		4117	1	1
	167-1LV	1	1		ND	1	1
*Klebsiella*	13	1	1				
	16	3	3	*Salmonella enterica*	1	10	10
	17	1	1				
	29	2	2	*Serratia marcescens*	ND	2	2
	37	2	2				
	39	1	1	*Enterobacter cloacae*	513	1	1
	45	2	2				
	54	1	1				
	101	1	1				
	147	1	1				
	218	1	1				
	231	1	1				
	268	1	1				
	307	2	2				
	485	1	1				
	690	1	1				
	716	1	1				
	870	1	1				
	882	1	1				
	987	1	1				
	2096	2	2				
	2944	1	1				
	3647	1	1				
	3702	2	2				
	4023	1	1				
	37-1LV	1	1				

## Data Availability

Original research data are available upon reasonable request to the research team, after permission from regulatory institutions. The raw sequence reads of the bacterial WGS are available at NCBI under BioProject accession number PRJNA1005941.

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
