# Peer review of "Epidemiology, Clinical, and Microbiological Characteristics of Multidrug-Resistant Gram-Negative Bacteremia in Qatar"

_antibiotics, 2024, doi:10.3390/antibiotics13040320_

Round 1
Reviewer 1 Report
Comments and Suggestions for Authors
Dear Editors of Antibiotics Journal
I trust you are well.
Herewith kindly receive my comments regarding the manuscript entitled “Epidemiology, clinical and microbiological characteristics of MDR Gram-negative bacteremia in Qatar”.
Kind regards

Moderate editing of the English language is required.
Author Response
Thank you for taking the time and effort to review our manuscript.
We addressed most raised points, especially the observed high susceptibility rates for carbapenems and amikacin. We also modified Figure 1 both at results and discussion as well as added detailed explanatory caption.

Reviewer 2 Report
Comments and Suggestions for Authors
In this study, the authors performed extensive research, focusing on one of the major issues globally. Antimicrobial resistance is a global threat affecting all the One Health components, including humans, animals, and environments. The present study evaluated the epidemiology, clinical, and microbiological characteristics of bloodstream infections in humans in Qatar. They vastly focused on Gram-negative bacteria. I must appreciate their research!
However, I found a few major issues, especially antibiotic selection. The authors should recheck it. Also, please find my comments below:
Title
Line 3: Please provide the full form of “MDR” here.
Abstract
Line 30: Please write "Escherichia coli, Klebsiella pneumoniae” instead of “E. coli, K. pneumoniae.” The scientific names of any organisms should be fully formed at their first use. If needed, please correct them throughout the manuscript.
Line 30-31: Please provide the prevalence of these MDR organisms here.
Keywords
Line 39: You don’t need to mention the abbreviations in the keywords section.
Introduction
Line 82: “aims” should be “aimed.”
M+M
Line 134-136: How did the authors perform multivariate logistic regression analyses? How did they create a logistic regression model? How did they select variables to create the regression model? Did they perform a collinearity test to do that? If yes, how? If not, please clarify this.
Results
Table 3:
- I was just wondering why you tested Pseudomonas aeruginosa against several antibiotics, e.g., ampicillin, amoxicillin-clavulanic acid, ceftriaxone, ertapenem, trimethoprim, and tigecycline. P. aeruginosa isolates show intrinsic resistance, which means they are inherently resistant to these antibiotics. You don’t need to perform AST to say that! You performed the almost perfect test as you found that these isolates are 100% resistant to these antibiotics, except you found 91.7% of isolates were resistant to tigecycline, which should be 100%! Please perform the AST for this antibiotic. However, I suggest removing these data from the manuscript!!!
- Klebsiella pneumoniae shows intrinsic resistance to ampicillin.
- Enterobacter cloacae are intrinsically resistant to ampicillin, amoxicillin-clavulanic acid (your result is questionable), cefazolin, cefoxitin, and cephalothin.
- Serratia marcescens exhibit intrinsic resistance to ampicillin, amoxicillin-clavulanic acid, cefazolin, cephalothin, cefuroxime, cefoxitin, colistin, polymyxin, and nitrofurantoin.
- With these intrinsic resistance data, you cannot confirm their multidrug resistance profiles.
Discussion
- You mentioned Table numbers in the discussion section. I think you don’t need to mention them again! If you think you must mention them in the discussion section, please clarify it.
- You should mention or discuss the intrinsic resistance of the bacteria in the discussion section. You can also do it in the introduction section.
- Line 271: I was just wondering why you used Figure 1 in the discussion section instead of the results section. Please clarify.
Author Response
VXXXXXXXXXXXXXXXXX
Thank you for taking the time and effort to review our manuscript.
We addressed most raised points specially table 3 which was an oversight from our part not to highlight intrinsic resistance of classical pathogens to certain antimicrobials. The results were collated from the Phoenix system and interpreted accordingly. We removed AST interpretations and replaced them with NA (non-applicable) to show designated antimicrobials are ineffective against specified pathogens, we hope that is satisfactory and not misleading to others.
XXCV

Round 2
Reviewer 2 Report
Comments and Suggestions for Authors
The authors addressed most of my comments adequately. However, please check whether the title of the manuscript should have an abbreviation.
Also, I have found some major issues!
- Table 3: The number of isolates and their percentages are not correct. Please check it properly and correct it. Also, please check all the isolates' numbers and percentages throughout the manuscript.
- I have another concern about the number of multidrug-resistant isolates. Did you check if all the 429 isolates were MDR in nature after removing those intrinsic resistance data? Please check it properly. If needed, please correct it properly throughout the manuscript. Also, I need confirmation for that. Your title of the study depends on this MDR characteristics.
Author Response
|
Thanks, you for your feedback and comments . We fully understand parts of the concerns which stems from the layout of the study . We identified 429 episodes of MDR GNB bacteraemia (important for epidemiology), but we only had 245 records for microbiological and clinical characteristics (for some internal logistical reasons), that why appear discrepant . We tried our best to be transparent by creating a study flow chart as well as added captions to tables to remove any ambiguity . For the MDR definitions, we included an elaborated responses in the attached file. |

Round 3
Reviewer 2 Report
Comments and Suggestions for Authors
Still, I have confusion about them. Please check my detailed comments below:
Table 3: The authors might not understand my comment. The number of isolates against different antibiotics is not consistent, e.g., in the case of E. coli, 159 to amikacin, 158 to cefoxitin, 156 to cefazolin, etc. Please check it for other bacteria and correct them properly!
MDR issue: I just need to be confirmed that all the 245 (or 429) isolates were MDR after omitting intrinsic resistance data. I mean, did you check all the isolates were resistant to at least three antimicrobial classes after removing their intrinsic resistance profiles?
I checked your flowchart. There were 429 isolates, right? You confirmed them using different antibiotics, includuing intrinsic resistance data, right?. But if you remove this data, the MDR profiles might be changed. But if it is not changed, I need to be confirmed that all the 429 isolates (or 245) are MDR till now.
Author Response
Thanks, you for the raised points.
- Now we understand the raised point of Table 3 : we agree the numbers did not tally so were reviewed (mainly for E.coli which should total 159 as indicated ) . There were issues interpreting cipro and levo breakpoint so were left blank at the excel sheet not captured during analysis. Now corrected them as highlighted .
- The MDR definitions were based beforehand on commonly tested antimicrobials for related pathogens . The initial collated results were from the automated system which should’ve been censored for pathogens related intrinsically resistant antimicrobial agents.
